# Sublytic gasdermin-D pores captured in atomistic molecular simulations

Stefan L Schaefer[1], Gerhard Hummer[1,2]*

[1]Department of Theoretical Biophysics, Max Planck Institute of Biophysics, Frankfurt am Main, Germany; [2]Institute of Biophysics, Goethe University Frankfurt, Frankfurt am Main, Germany

**Abstract** Gasdermin-D (GSDMD) is the ultimate effector of pyroptosis, a form of programmed cell death associated with pathogen invasion and inflammation. After proteolytic cleavage by caspases, the GSDMD N-terminal domain (GSDMD[NT]) assembles on the inner leaflet of the plasma membrane and induces the formation of membrane pores. We use atomistic molecular dynamics simulations to study GSDMD[NT] monomers, oligomers, and rings in an asymmetric plasma membrane mimetic. We identify distinct interaction motifs of GSDMD[NT] with phosphatidylinositol-4,5-bisphosphate ($PI(4,5)P_2$) and phosphatidylserine (PS) headgroups and describe their conformational dependence. Oligomers are stabilized by shared lipid binding sites between neighboring monomers acting akin to double-sided tape. We show that already small GSDMD[NT] oligomers support stable, water-filled, and ion-conducting membrane pores bounded by curled beta-sheets. In large-scale simulations, we resolve the process of pore formation from GSDMD[NT] arcs and lipid efflux from partial rings. We find that high-order GSDMD[NT] oligomers can crack under the line tension of 86 pN created by an open membrane edge to form the slit pores or closed GSDMD[NT] rings seen in atomic force microscopy experiments. Our simulations provide a detailed view of key steps in GSDMD[NT]-induced plasma membrane pore formation, including sublytic pores that explain nonselective ion flux during early pyroptosis.

*For correspondence:
gerhard.hummer@biophys.mpg.de

**Competing interest:** The authors declare that no competing interests exist.

## Editor's evaluation

This article will be of interest to cell biologists, structural biologists, and biophysicists studying programmed cell death, membrane transport, and protein-lipid interactions. The simulation data presented offers atomistic detail of how gasdermin-D N-terminal domains assemble on the plasma membrane and trigger the formation of membrane pores which lead to pyroptosis. The study is well-designed and the resulting data are rigorously analyzed.

## Introduction

Pyroptosis is a recently discovered form of regulated cell death that leads to lysis of the affected cell and to the release of damage-associated molecular patterns (DAMPs) such as mature IL-1β and IL-18 (*He et al., 2015*; *Tsuchiya et al., 2021*; *Xia et al., 2021*; *Xie et al., 2022*). Pyroptosis is tightly regulated by the assembly of canonical or non-canonical inflammasomes and the activation or presence of certain caspases, granzymes or pathogenic proteases (*Shi et al., 2015*; *Kayagaki et al., 2015*; *Liu et al., 2021*; *Deng et al., 2022*; *Wen et al., 2021*). In turn, these proteases activate gasdermins by cleaving off their C-terminal domains, which exposes a basic surface on the N-terminal domain (*Liu et al., 2021*; *Xia et al., 2021*). The N-terminal domain then binds to the plasma membrane, oligomerizes, and inserts a β-sheet into the membrane, similar to bacterial β pore-forming toxins (*Dal Peraro and van der Goot, 2016*) though acting from the inner leaflet. The hydrophobic face of the β-sheet

anchors the sheet into the membrane and its hydrophilic face facilitates the opening of an approximately 20 nm wide membrane pore (*Ruan et al., 2018*; *Xia et al., 2021*). In addition to cytokine release, the opening of the pore in most cases leads to an osmotic shock that ultimately disrupts the integrity of the cell (*Fink and Cookson, 2006*). Gasdermins play a crucial role in the innate immune response to pathogen infection. Altered activation of pyroptosis has also been associated with various types of cancers and cancer treatments (*Liu et al., 2021*; *Wu et al., 2021*). A detailed, molecular-level understanding of gasdermin action is thus biomedically relevant (*Ryder et al., 2022*).

Gasdermin-D (GSDMD) is the best-characterized gasdermin of the six human isoforms. It is expressed in many tissues, including cells of the gastrointestinal system, the circulatory system, the skin, the lung, and many immune cells (*Broz et al., 2020*). Its central role in infection response makes GSDMD a target of therapeutic applications (*Liu et al., 2021*). After proteolytic cleavage, the GSDMD N-terminal domain (GSDMD$^{NT}$) binds specifically to negatively charged lipids of the inner leaflet of the plasma membrane, where it forms β-pores comprising around 30 subunits (*Aglietti et al., 2016*; *Ding et al., 2016*; *Liu et al., 2016*; *Xia et al., 2021*; *Mulvihill et al., 2018*).

The exact order and mechanism with which GSDMD$^{NT}$ binds the plasma membrane, oligomerizes, and spontaneously inserts its transmembrane β-hairpins into the membrane is not fully understood. Cryo-electron microscopy (cryo-EM) studies resolved the pore structures of mouse gasdermin-A3 (GSDMA3) (*Ruan et al., 2018*) and human GSDMD (*Xia et al., 2021*), respectively, and identified the density of a fully assembled prepore ring stacked opposite the pore structure and separated by detergent. Very recently, atomic force microscopy (AFM) imaging resolved circular assemblies that could be washed off the supported bilayer (*Mari et al., 2022*). These results promote the idea that GSDMD$^{NT}$ forms complete rings before folding and inserting its β-sheet. On the other hand, AFM experiments of growing assemblies (*Mulvihill et al., 2018*) and the observation of nonselective ion influx and efflux in the early phases of pyroptosis (*de Vasconcelos et al., 2019*; *Chen et al., 2016*) favor an alternative pore assembly pathway, in which GSDMD$^{NT}$ first forms membrane-inserted arcs and slits, which can then grow over time to build full circular pores (*Rühl and Broz, 2022*).

Here, we use multi-microsecond atomistic molecular dynamics (MD) simulations to study the lipid interactions, dynamics, and structural stability of differently sized GSDMD$^{NT}$ oligomers and rings in prepore and pore conformation. For our simulations, we use a realistic asymmetric plasma membrane mimetic. The simulations give us a detailed view of lipid binding, which is critically important to target GSDMD$^{NT}$ to the inner leaflet of the plasma membrane and may play a role in inducing the transition from prepore to pore conformation. The simulations also allow us to watch in real time as lipids recede from membrane-inserted GSDMD$^{NT}$ oligomers to form stable membrane pores with diameters from 1 to 20 nm. In turn, we show that the resulting membrane forces impact the formation and relative stability of GSDMD$^{NT}$ arc, slit, and circular ring structures, as seen in experiments. We conclude by contrasting the different pore formation pathways and raising open questions.

## Results

### GSDMD$^{NT}$ binds to acidic lipids in the plasma membrane

GSDMD$^{NT}$ strongly interacted with acidic lipids in all our simulations of the prepore and pore conformations. We observed noticeable enrichment of lipids with acidic phosphatidylinositol-4,5-bisphosphate (PI(4,5)P$_2$) and phosphatidylserine (PS) headgroups around the protein already during the equilibration phase. In comparison to zwitterionic lipids (phosphatidylethanolamine (PE) and phosphatidylcholine (PC)), anionic lipids interacted more strongly with GSDMD$^{NT}$ than what would naïvely be expected based on their abundance in the inner leaflet (*Figure 1—figure supplement 1*). In the production phase of the MD simulations, the enrichment of the anionic PS and PI(4,5)P$_2$ lipids continued, resulting in the formation of distinct clusters at the protein underside, stabilized by abundant basic amino acids.

Distinct sites were occupied with high consistency. The β1–β2 loop (residues 42–55; numbering as in the cryo-EM structure by *Xia et al., 2021*) emerged as a focal point of interactions with anionic lipids (*Figure 1A and B*). Its aromatic residues W48, F49, W50, and Y54 anchored the loop deeply into the upper membrane leaflet. Anionic lipids clustered at the interface of the β1–β2 loop with the α1 helix (dominated by interactions with R10, K51, R53), the α3 helix (dominated by interactions with K43, R53, K55, R153), and the disordered C-terminus (dominated by interactions with K51, K235, K236, R238). An additional binding site for anionic lipids between α1 and β7 was stabilized by

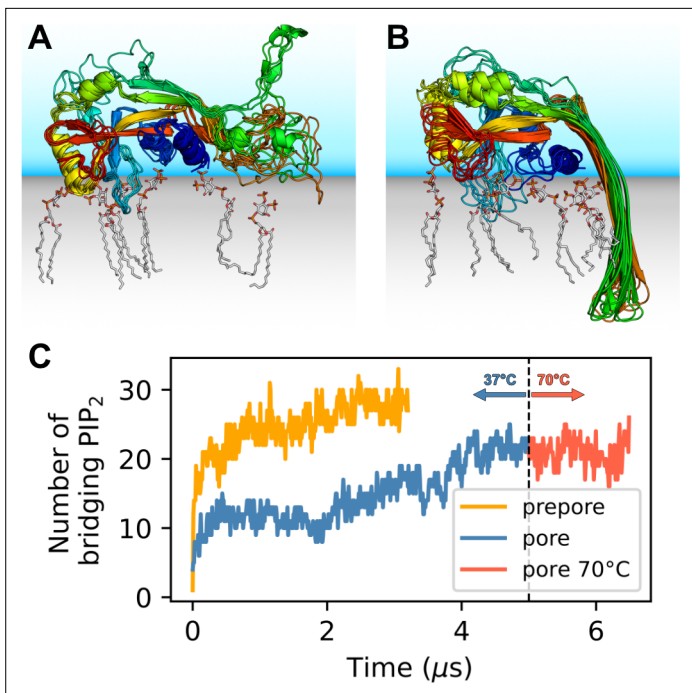

**Figure 1.** GSDMD[NT] interacts tightly with anionic lipids. Overlay of six representative PI(4,5)P₂ bound poses of the prepore monomer (**A**) and the 33-mer GSDMD[NT] ring in pore conformation (**B**). GSDMD[NT] is shown in cartoon representation and colored using a rainbow spectrum from blue (N-terminus) to red (C-terminus). The β1–β2 loop is colored in cyan, the α1 helix in dark blue, the α3 helix in yellow, and the C-terminus in red. PI(4,5)P₂ is shown in grey licorice representation with orange phosphorus and red oxygen atoms. Hydrogen atoms are not shown for clarity. The membrane and solvent are schematically shown with gray and blue shades, respectively. (**C**) Number of PI(4,5)P₂ molecules that interact simultaneously with two subunits of the prepore (orange) and pore (blue) 33-mer rings. After 5 μs at 37°C, the pore simulation was continued for 1.5 μs at 70 °C (red).

The online version of this article includes the following source data and figure supplement(s) for figure 1:

**Source data 1.** Source data for *Figure 1C*.

**Figure supplement 1.** Change in the number of inner leaflet lipids, whose headgroups interact with at least one heavy atom of the 33-mer pore, in absolute counts (top) and normalized by the number of lipids of each lipid species in the inner leaflet (bottom).

**Figure supplement 1—source data 1.** Source data for Figure 1—figure supplement 1.

interactions with R7, R10, R11, and R178. In around two thirds of interfaces between neighboring subunits in the 33-mer pore, we found a PI(4,5)P₂ headgroup that interacted with both neighbors at the same time (*Figure 1C*). The flexible, unfolded hairpins of the prepore 33-mer appear to stabilize bridging PI(4,5)P₂ between neighboring subunits.

The lipid interactions in the prepore monomer and the 33-mer pore differ in three notable aspects. (1) In the prepore monomer, the loops eventually forming the membrane spanning β-sheet in the pore conformation remained mostly unfolded and resided on the membrane surface with only few residues penetrating the interface (*Figure 1A*, green and orange). Their basic amino acids faced the membrane to form lipid contacts. By contrast, in the pore conformation, only R178, R174, and K204 bound PI(4,5)P₂, whereas the side chains of the other basic hairpin residues pointed towards the water-filled pore (K177, K203) or stabilized the tips of the sheet in the extracellular leaflet (K103, R183). In the pore conformation, a single PI(4,5)P₂ bound to R7, R174, and R178 simultaneously in some instances. (2) In the pore conformation, the α1 helices (*Figure 1B*, dark blue) formed a continuous belt lying flat on the membrane. By contrast, in the prepore monomer the α1 helix was tilted at an angle of ≈30° with respect to the membrane plane, with the N-terminus pointing towards the membrane (*Figure 1A*, dark blue). (3) Whereas the α3 helix (*Figure 1A and B*, yellow) was lifted off the membrane in the pore

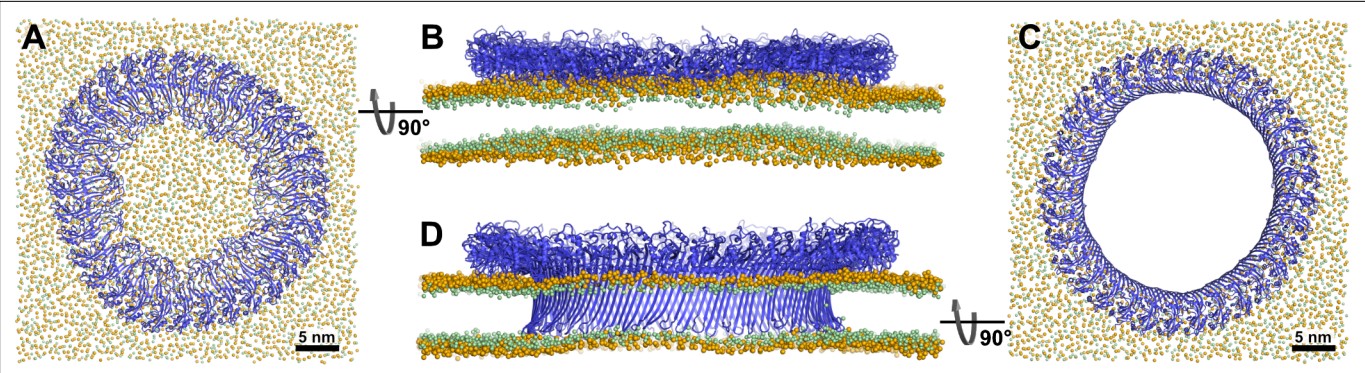

**Figure 2.** Atomistic MD simulations of GSDMD[NT] 33-mer rings in prepore and pore conformation. Prepore (A,B; after 3.5 µs of MD) and pore rings (C,D; after 5 µs) are viewed from the top (**A,C**) and side (**B,D**). GSDMD[NT] is shown in blue cartoon representation, lipid headgroup phosphates and glycerol oxygens are shown as orange and green spheres, respectively. Water, ions, and lipid tails are not shown for clarity. The membrane under the prepore ring (**A,B**) is continuous but visibly bent upwards into the ring (**B**). In the pore conformation (**C,D**), lipids are absent from the central pore, which is lined by a continuous, membrane spanning β-barrel.

The online version of this article includes the following figure supplement(s) for figure 2:

**Figure supplement 1.** Snapshots of prepore 33-mer on a larger membrane patch and of prepore 3, 5, and 16-mer on one membrane patch.

conformation, in the prepore monomer its C-terminal residues lay directly on the membrane interface. This created a lipid binding site between α3 around the C-terminus and residues K145, Q149, and R151, which is unique to the monomeric prepore conformation. R153 formed frequent contacts to acidic lipid headgroups in both conformations.

## Prepore GSDMD[NT] arcs and rings deform lipid membrane

To gain insight into the dynamics and stability of GSDMD[NT] ring assemblies, we performed simulations of the full 33-mer assemblies, starting separate simulations from its proposed prepore conformation and from its resolved pore conformation (*Xia et al., 2021*). In the prepore conformation, the 33-mer complex maintained a nearly circular shape and remained tightly bound to the intracellular leaflet of the plasma membrane for the entirety of the 3.5 µs simulation (*Figure 2A*). Within the first nanoseconds of the production simulation, its originally folded β-hairpins unfolded and the ring deformed the membrane upwards into a crown shape (*Figure 2B*). To assess whether this upwards bending is affected by the limited size of the membrane, we performed another simulation of the circular prepore structure on a larger membrane (46×46 nm²) for 2.2 µs. There, the upwards bending caused by GSDMD[NT] was even more pronounced (*Figure 2—figure supplement 1B*). Noticeably, however, in this larger system, broken inter-subunit contacts between neighboring GSDMD[NT] globular domains resulted in distortions of the ring shape (*Figure 2—figure supplement 1A*). It is conceivable that the local cracks in the ring were caused by the stress resulting from more pronounced membrane deformations in the system with a larger membrane patch.

In addition to the full 33-mer prepore, we also performed simulations of membrane-adhered 3, 5, and 16-mers. The oligomers remained stable in the simulated time and kept their original arc shape. The membrane under the 16-mer arc was bent upwards, albeit not to the extent seen under the 33-mer prepore ring (*Figure 2—figure supplement 1C*).

## GSDMD[NT] 33-mer rings support stable membrane pores

In the pore conformation, GSDMD[NT] stabilized a 21.6 nm wide, water-filled pore with a fully intact β-sheet lining its side (*Figure 2C and D*). During the entire simulation (5 µs), the membrane remained flat (*Figure 2D*), which is in contrast to the prepore ring. However, around the tips of the inserted hairpins the membrane was thinned because the hairpins are not long enough to reach fully across the membrane to the lipid headgroups of the extracellular leaflet. Nevertheless, the overall topology only deviated minimally from its initial perfectly circular shape and no gaps in the ring of globular domains opened up in 5 µs of MD.

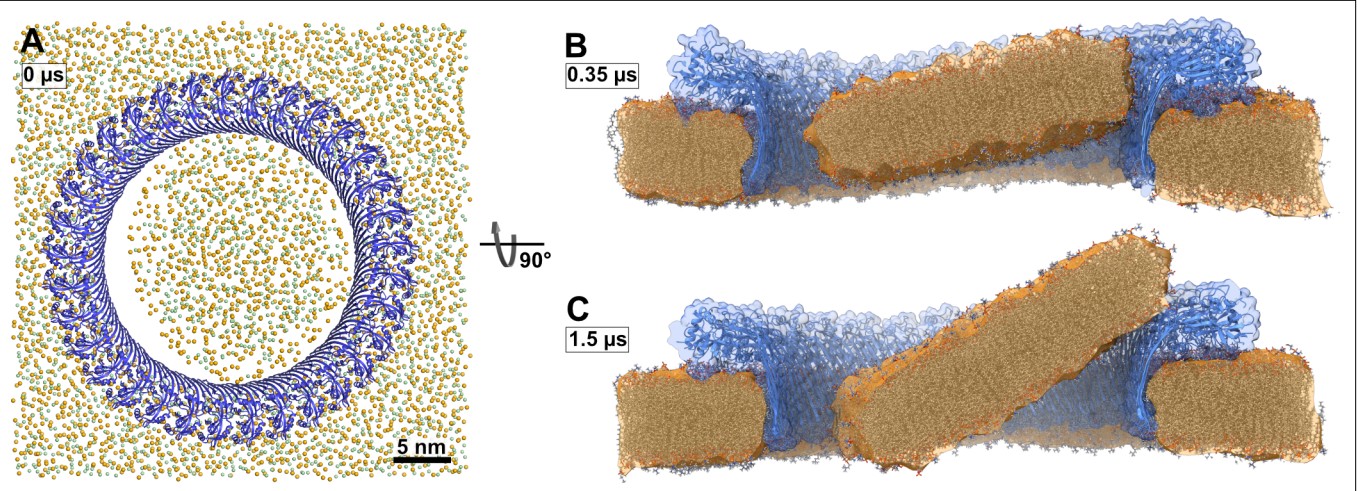

**Figure 3.** Atomistic MD simulations of GSDMD[NT] 33-mer rings in pore conformation and filled initially with lipids. (**A**) Top view after equilibration. (**B,C**) Side views after 0.35 µs (**B**) and after 1.5 µs (**C**) of MD simulation shown as section through the pore center. The GSDMD[NT] backbones are shown in blue cartoon representation. In (**A**), lipid headgroup phosphates and glycerol oxygen atoms are shown as orange and green spheres, respectively. Water, ions, and lipid tails are not shown for clarity. In the sections (**B,C**), lipid tails are shown in full licorice representation. The outlines of the protein and membrane are represented as transparent surfaces.

In addition to the open and water-filled pore, we also simulated a pore that was filled initially with a lipid bilayer to mimic the events following a presumed concerted β-sheet insertion of a prepore ring. In this simulation, the lipid patch blocking the pore detached from the hydrophilic inside of the membrane-spanning β-sheet already during the equilibration phase (*Figure 3A*). A bicelle-like plug formed that then started to tilt. As it tilted, the plug moved slowly in the direction of the intracellular space along the protein surface (*Figure 3B and C*). On the timescale of the simulation, the plug remained planar. Due to its slow diffusion, we could not capture the full release of the plug within the simulated time. Nevertheless, the detachment, tilt, and displacement of the plug together give a clear indication of the pathway to pore opening induced by a 'cookie-cutter-like' concerted membrane insertion of the β-sheet.

## Small oligomers of membrane-inserted GSDMD[NT] support stable membrane pores

To test whether small oligomers can remain stably inserted in the membrane, we performed simulations of a membrane inserted monomer and of different-size oligomers (2, 3, 5, 8, and 10-mer), starting from arc-like segments taken out of the circular 33-mer cryo-EM structure in pore conformation (*Xia et al., 2021*). Independent of the number of subunits, all GSDMD[NT] systems, including the monomer, remained stably inserted in the membrane for the entirety of our simulations (*Figure 4A–F*). On the pore-facing side of the inserted β-sheet, this led to disruptions of the membrane integrity, because the hydrophilic residues on this side drew water and phospholipid headgroup moieties into the hydrophobic membrane core region. While the water chains along the small sheet of monomeric GSDMD[NT] were regularly interrupted, all oligomeric systems maintained a continuous water column on the pore-facing side of the inserted β-sheet. Sodium and chloride ions permeated in both directions through the water-filled oligomeric pores (*Figure 4G*). The number of permeation events increased with oligomer size.

In systems with two or more GSDMD[NT], we found that the β-sheet curled up to form pores (*Figure 4B–F*). Whereas the edge of the sheet formed by the β3 strand stayed normal to the membrane plane, the other end of the sheet formed by the β7–β8 hairpin tilted upwards to an angle of ≈55° and almost crossed the membrane plane (*Figure 4C and E*, front view). In the resulting pores of oval shape, the bent β7–β8 hairpin coats one of the highly curved narrow membrane edges.

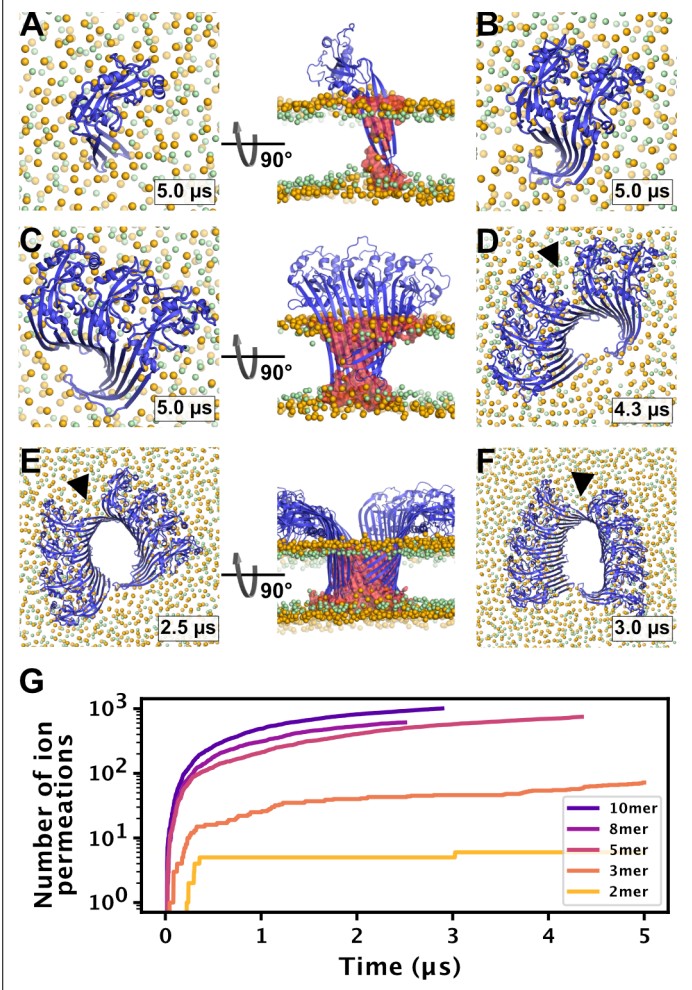

**Figure 4.** MD simulations of small GSDMD[NT] oligomers. GSDMD[NT] monomer (**A**), dimer (**B**), trimer (**C**), pentamer (**D**), octamer (**E**) and decamer (**F**) remain membrane inserted for the full duration of the respective MD simulations. The β-sheets of 2, 3, 5, 8, and 10-mers coil up into small membrane pores filled with water (water inside the pore shown as red volume in the right panels of A,C,E). The GSDMD[NT] backbones are shown in blue cartoon representation. Lipid headgroup phosphates and glycerol oxygens are shown as orange and green spheres, respectively. Water, ions, and lipid tails are not shown for clarity except in the right panels of A,C,E. Black triangles indicate sites where the arc had cracked. (**G**) Cumulative sodium and chloride ion permeation events during the simulations. No ions permeated the membrane in the monomer simulation.

The online version of this article includes the following source data for figure 4:

**Source data 1.** Source data for *Figure 4G*.

## High membrane edge tension drives the formation of slit and ring shaped pores

We also performed atomistic multi-microsecond MD simulations of larger membrane inserted oligomers with 16 and 27 GSDMD[NT], respectively. In the GSDMD[NT] 16-mer and 27-mer simulations (*Figure 5A and B*), the plasma membrane lipids receded quickly from the hydrophilic face of the β-sheet, often already during the equilibration steps. In concert, water flowed into the space vacated by the lipids along the sheet. On the side of the receding membrane, phospholipid headgroups wrapped around the now open membrane edge to shield the otherwise exposed hydrophobic membrane core from water. Across the open edge, the intracellular and the extracellular membrane leaflets could exchange lipids.

On a much longer timescale in the simulations, the length of this open membrane edge shortened. First, the edge straightened by receding further from the interior of the pore; then, the open

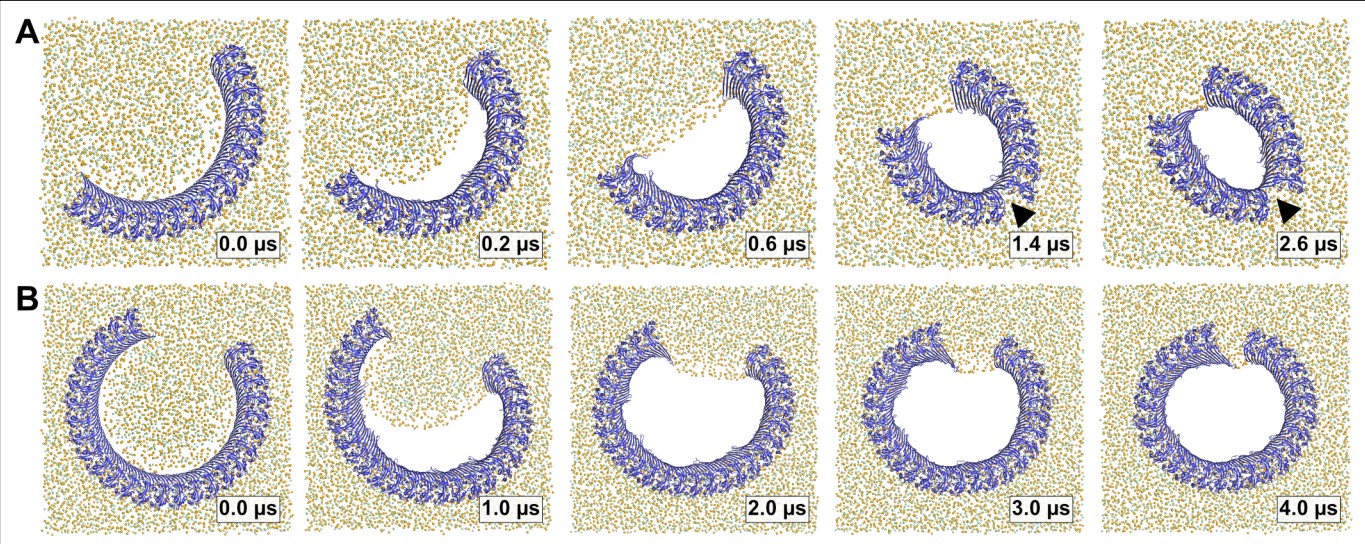

**Figure 5.** Arc-shaped GSDMD$^{NT}$ oligomers transition into slit or ring-shaped membrane pores. Top views of GSDMD$^{NT}$ arcs comprising 16 (**A**) and 27 (**B**) subunits in pore conformation along MD simulation trajectories (time points indicated) show phospholipid headgroups (orange spheres) and cholesterol oxygens (green spheres) receding from the inserted β-sheet, before the open protein edges approach each other and close into slit-shaped (**A**) or ring-shaped (**B**) pores. Water, ions, and lipid tails are not shown for clarity.

The online version of this article includes the following video and figure supplement(s) for figure 5:

**Figure supplement 1.** Time lapse images of pore formation from 5–10-meric arcs in the plasma membrane and a 16-meric arc in a pure DOPC membrane.

**Figure 5—video 1.** Formation of a slit-like pore from a membrane inserted 16-meric arc.

https://elifesciences.org/articles/81432/figures#fig5video1

**Figure 5—video 2.** Formation of a ring-like pore from a membrane inserted 27-meric arc.

https://elifesciences.org/articles/81432/figures#fig5video2

ends of the arc-shaped GSDMD$^{NT}$ multimers were gradually pulled together. In case of the 27-mer arc, this contraction resulted in the formation of a circular pore with a diameter of ≈17 nm within 4 µs (*Figure 5B*, *Figure 5—video 2*). By contrast, in all simulations of oligomers comprising between 5 and 16 subunits, the initial arcs cracked at the center as the membrane edge shortened and contacts between the globular domains of neighboring subunits were lost (*Table 1*, *Figure 5—figure supplement 1*). The 16-mer cracked after ≈0.9 µs, which resulted in the formation of a kink in the arc (*Figure 5A*, *Figure 5—video 1*). As the open ends of the two sub-arcs further contracted, a slit-shaped membrane pore formed. Importantly, despite the loss of contacts between the globular domains, the inserted β-sheets of all cracked arcs stayed fully intact, even in the highly curved cracked region of the pores.

## Membrane edge tension exerts large force on gasdermin arcs

**Table 1.** Summary of arc cracking events.

| System size | Cracked subunit interfaces | Time of cracking [µs] |
|---|---|---|
| 5 | 2,3 | 2.91 |
| 8 | 4,5 | 0.23 |
| 10 | 5,6 | 0.74 |
| 16 | 8,9 | 0.89 |
| *16 | 6,7; 9,10; 11,12 | 0.4; 0.26; 0.27 |

*Simulated in pure DOPC membrane.

We quantified the force acting on the open protein edges by determining the edge tension for our plasma membrane mimetic. From a simulation of a system without protein and with two plasma membrane edges that could not close under the condition of fixed box area, we determined the membrane edge tension per unit length of open edge as $\gamma = 86.4 \pm 3.9$ pN for our plasma membrane mimetic. This edge tension is the force exerted onto the ends of the gasdermin arcs once the edge has straightened. On a molecular scale, this force is large, amounting to a drop in free energy by 21 $k_B T$ for a 1 nm edge shortening.

### Gasdermin-D arc in DOPC membrane closes into circular pore

For reference, we also performed a simulation of a GSDMD$^{NT}$ 16-mer in a pure 1,2-dioleoyl-sn-glyc ero-3-phosphocholine (DOPC) membrane. For DOPC, a lower membrane edge tension of 44.3±2.1 pN (*West et al., 2013*) has been reported. Compared to the plasma membrane mimetic, the lipids receded more rapidly from the inserted β-barrel to form a pore (within 400 ns, *Figure 5—figure supplement 1*). As the membrane edge contracted, the GSDMD$^{NT}$ arc cracked in three positions to form a nearly circular pore rather than the slit-shaped pore seen in the plasma membrane mimetic.

## Discussion

### Shared lipid binding sites could promote oligomerization

Lipid interactions play an important role in the function of pore-forming proteins (*Hodel et al., 2021*). *Xia et al., 2021* identified three basic patches and with the help of mutational studies confirmed that they partake in the recognition of acidic lipid headgroups: the basic α1-helix (basic patch (BP) 1, 'thumb'), the loop between β-strands 1 and 2 (β1–β2 loop, BP2, 'wrist'), and two basic residues (R174, K204) of β-strands 7 and 8, respectively. In the MD simulations, we found that these basic patches do not act independently of each other. In particular, the flexible β1–β2 loop shares interaction sites with all of the other arginine and lysine rich sites. Further, we were able to specify several residues that we propose to fulfil key functions in GSDMD$^{NT}$ membrane binding.

In addition, our simulations revealed that GSDMD$^{NT}$ binds the plasma membrane differently, depending on whether it is in its monomeric prepore conformation or its oligomeric pore confor- mation. Differences in the strength of these interactions may drive the reorientation of GSDMD$^{NT}$ on the membrane during the transition from prepore to pore conformation. The described differ- ences also point to certain residues involved in membrane binding, but not pore formation, such as K145, Q149, and R151. In variants with K145 and R151 mutated to alanine, oligomerization and pyroptosis are compromised, as is localization to the detergent phase in lysed cells (*Liu et al., 2016*). Furthermore, while entropically unfavorable, binding of a single PI(4,5)P$_2$ to R7, R174, and R178 at the same time may kinetically trap these three residues in a distinct pore-like orientation long enough to facilitate the folding of the β-hairpin. This may also explain the preference of gasdermins to form pores in membranes containing multivalent acidic lipids such as PI(4,5)P$_2$, phosphatidylinositol-3,4,5- triphosphate (PI(3,4,5)P$_3$), and cardiolipin (*Ding et al., 2016*). Finally, we frequently observed PI(4,5)P$_2$ bridging the interface of two neighboring subunits via their lateral binding sites (*Figure 1C*).

Even if the bridging lipids (*Muller et al., 2019*) acting as 'double-sided tapes' may not confer high mechanical strength to the multimerization interface, they should facilitate oligomerization by attracting membrane-bound gasdermins and by helping to orientate them with respect to each other. It is therefore also interesting to see that the 33-mer in prepore conformation is even more intercon- nected by PI(4,5)P$_2$ than the 33-mer in pore conformation (*Figure 1C*). These lipid-mediated interac- tions should facilitate prepore assembly on the plasma membrane, which is rich in PI(4,5)P$_2$. Combined with our observation of joint binding sites within one subunit, this mechanism could provide an addi- tional explanation for the preference of GSDMD$^{NT}$ for multivalent acidic lipids (*Ding et al., 2016*) and for the recent observation that it shows less diverse pore conformations in PI(4,5)P$_2$ or PI(3,4,5)P$_3$ rich membranes than in pure PE/PC membranes (*Santa Cruz Garcia et al., 2022*). Similar to the recruit- ment of PI(4,5)P$_2$ by GSDMD$^{NT}$, prepore oligomers of pneumolysin (PLY), a cholesterol dependent cytolysin (CDC), have been shown to recruit cholesterol, which locally enhanced lipid order (*Faraj et al., 2020*).

### Pores may grow by fusion of small membrane inserted segments

In earlier work, we found that the membrane β-sheet of monomeric PLY was pushed out of the membrane within 1 µs in one of two atomistic replica simulations (*Vögele et al., 2019*). It was there- fore surprising to us that the β-hairpins of monomeric and dimeric GSDMD$^{NT}$ stayed stably membrane inserted for the entirety of our 5 µs simulations. Combined with our observation of stable prepore oligomers and the identification of stable, presumably inserted, arcs of the CDC suilysin comprising as few as five subunits (*Leung et al., 2014*), this finding opens up the possibility that GSDMD$^{NT}$ inserts its β-sheet already from such small oligomers. In contrast to PLY, GSDMD$^{NT}$ was shown to permit growth of inserted arcs (*Mulvihill et al., 2018*) and we believe that such stably inserted small oligomers could

diffuse together to form slit and ring-shaped pores (*Leung et al., 2014*). In addition, the fact that the pores formed by small oligomers already permit water and ion conduction across the membrane is perfectly in line with observed ion flux and size exclusion in the early phases of pyroptosis (*de Vasconcelos et al., 2019*; *Chen et al., 2016*; *Rühl and Broz, 2022*). Sterically, release of IL-1 family cytokines (*Kato et al., 2003*; *Finzel et al., 1989*) will require pores of at least 10 GSDMD$^{NT}$ subunits.

Furthermore, we occasionally found the interface between globular domains of neighboring subunits to break under stress. By contrast, the membrane inserted β-sheet always stayed intact, even in the highly curved kink region of slit-shaped pores. This remarkable stability of the β-sheet and adaption to high curvature suggests that β-sheet formation may stabilize GSDMD$^{NT}$ oligomers in pore conformation.

In AFM experiments, *Mulvihill et al., 2018* identified an abundance of membrane inserted slit-shaped pores. Recently, *Santa Cruz Garcia et al., 2022* discovered that GSDMD$^{NT}$ pores may be able to dynamically open and close and they discuss that slit-shaped pores may provide the closed state to which open rings can collapse. With our 16-mer membrane inserted system, we identified a pathway that leads to the formation of pores similar in shape and size to these experimentally reported ones. We further show that slit-shaped pores can form spontaneously from arcs, driven by the tension along an open membrane edge.

Taken together, these results suggest that already monomers and small oligomers can support stable membrane insertion. Similar to the α pore-forming toxin ClyA, these smaller inserted oligomers may fuse with one another to build larger slit and finally ring-shaped pores (*Benke et al., 2015*). This assembly, however, will happen on much longer timescales than currently accessible with atomistic MD simulations.

Lastly, the atomistic resolution of our simulations ensures a realistic flexibility of the protein assemblies, which allowed the open edge caused by β7 and β8 to partly rotate out of the membrane. As a result, one edge of the β-sheet is partly exposed. It would therefore provide an excellent attachment point for the β-strands of non-inserted subunits at the membrane surface, and may guide them towards the growing assembly and, eventually, into the membrane. In this way, the upward-tilt of one β-sheet edge would drive sequential oligomer growth along this edge.

## Pore formation from prepore ring

From the above considerations, two competing assembly pathways emerge. High concentrations of prepore GSDMD$^{NT}$ on the membrane would promote assembly into full prepore rings before folding and inserting the continuous β-sheet. The observed upward bending of the membrane into the prepore ring inside is consistent with a pull on the domain that has to unravel to form the membrane inserted β-barrel. In a process similar to how antimicrobial peptides are believed to facilitate pore formation, locally confined adherence of many of the thus far unfolded hairpins would destabilize the membrane within the ring (*Flores-Romero et al., 2020*). The resulting kink in the membrane at the inner edge of the prepore ring (*Figure 2B* and *Figure 2—figure supplement 1B*) may facilitate the insertion of the β-strands. The instabilities we see in the globular domains of the circular prepore assembly are likely coupled to this strong membrane deformation.

## Bilayer and solvent composition influence pore formation

The lipid composition has a crucial effect on the assembly of pore forming proteins on the membrane and the subsequent insertion of their pore opening components (*Rojko and Anderluh, 2015*). Higher membrane fluidity should lead to easier insertion of the membrane penetrating components and facilitate the formation of smaller pores. By contrast, increased lipid order should facilitate the formation of larger pores.

Here, we used a complex, asymmetric membrane composition mimicking the plasma membrane (*Lorent et al., 2020*). The high cholesterol content makes the membrane comparably stiff, which in turn leads to slower diffusion and higher tension of the open membrane edge. As such, it differs from typical lipid mixtures that are commonly used for microscopic experiments of GSDMD (*Mulvihill et al., 2018*; *Hu et al., 2020*; *Xia et al., 2021*). For these experimental model membranes, we expect edge tensions about half of that of our plasma membrane mimetic (*Leomil et al., 2021*). By simulating the 16-mer arc structure also in pure DOPC, we could illustrate the potential effect of faster lipid

diffusion: despite the lower edge tension, small pores form faster and are more circular than in the more rigid plasma membrane.

As further complications, the plasma membrane of eukaryotic cells interacts with the cytoskeleton and the bilayers used in AFM experiments are usually supported on a solid surface. Indeed, single particle tracking, spectroscopic measurements, and MD studies showed substantially different lipid diffusion in the directly supported and the unsupported leaflet (*Schoch et al., 2018*; *Otosu and Yamaguchi, 2018*; *Roark and Feller, 2008*; *Koutsioubas, 2016*). We expect these interactions to impact the pore assembly pathways, in particular by lowering the membrane edge tension, an effect amplified by dissolved molecules and proteins that preferentially bind to the open edge, and by slowing the dynamics in the membrane. How strong these effects can be is impressively highlighted by experimental observations of certain detergents lowering the membrane edge tension by up to two orders of magnitude and by the fact that lipids purchased from different suppliers yield drastically different strengths of the edge tension, presumably due to impurities (*Karatekin et al., 2003*; *Puech et al., 2003*).

As an additional factor, arcs may be artificially stabilized on a crowded membrane by contacts to neighboring GSDMD[NT] oligomers (e.g., by forming stacked or interlocking arcs). Together, these effects may explain the observation that arcs are quite abundant in experiments on GSDMD[NT] and other β pore forming proteins (*Mulvihill et al., 2018*; *Mulvihill et al., 2015*; *Sborgi et al., 2016*; *Vögele et al., 2019*; *Leung et al., 2014*), yet above a certain size do not remain stable in our simulations.

## Differences between GSDMD and GSDMA3

A recent independent study of mouse GSDMA3 N-terminal domain (GSDMA3[NT]) (*Mari et al., 2022*) allows us to compare different gasdermins. In coarse-grained simulations, a 14-mer inserted GSDMA3[NT] was found to break up and transition to a slit-like pore (*Mari et al., 2022*), similar to what we observed here in atomistic MD simulations of a GSDMD[NT] 16-mer. Also consistent with our findings for GSDMD[NT], GSDMA3[NT] rings in pore conformation remained stable in atomistic MD simulations, whereas prepore rings proved comparably fragile and broke up (*Mari et al., 2022*).

As a crucial difference to GSDMD[NT], the small GSDMA3[NT] oligomer (*Mari et al., 2022*) did not form a membrane pore. Within 4 μs of atomistic MD, the membrane did not detach from the arc-shaped GSDMA3[NT] 7-mer to form a membrane edge (*Mari et al., 2022*). Missing the driving force for contraction, the arc remained stable without curling up. This is in stark contrast to our observations that water-filled and ion-conducting pores formed quickly for all oligomer sizes (*Figure 4*). As discussed above, the choice of membrane will likely impact the shape and stability of pores, and the missing sterols in the *E. coli* polar lipid extract (*Mari et al., 2022*) will have resulted in a less stiff membrane. However, a large difference in the hydrophilicity of the pore-facing residues is the most likely cause for the detachment of the lipid bilayer from GSDMD[NT] but not GSDMA3[NT]. On the Eisenberg hydrophobicity scale (*Eisenberg et al., 1982*), the GSDMD[NT] pore is 60% more hydrophilic than the GSDMA3[NT] pore (*Table 2*). Therefore, the difference between our results for GSDMD[NT] oligomers and those for a GSDMA3[NT] 7-mer (*Mari et al., 2022*) highlight how the physicochemical properties of the different gasdermins may be tuned to very specific membrane environments, cellular contexts or pore formation pathways. Based on the marked differences in hydrophilicity of the pore-facing side of the inserted β-sheets, it is tempting to speculate that—unlike GSDMA3[NT]—GSDMD[NT] allows

**Table 2.** Eisenberg hydrophobicity scores (*Eisenberg et al., 1982*) of the pore facing residues of human GSDMD and mouse GSDMA3 in kcal mol$^{-1}$.

| Structural element | human GSDMD | | mouse GSDMA3 | |
|---|---|---|---|---|
| | Pore facing residues | ΣEisenberg hydrophobicity | Pore facing residues | ΣEisenberg hydrophobicity |
| β3 | ADQQSE | −2.73 | MDQQLE | −1.93 |
| β5 | KAGASS | −0.96 | TKKTGS | −2.66 |
| β7 | TKESRS | −4.18 | TNNISP | −1.06 |
| β8 | QEQHSK | −3.76 | LGQSNN | −1.54 |
| Σ full sheet | | −11.63 | | −7.19 |

the formation of sublytic pores that facilitate nonselective ion flux and thus dissipate the membrane potential, as seen in our simulations (*Figure 4*) and in experiment (*de Vasconcelos et al., 2019*; *Rühl and Broz, 2022*; *Chen et al., 2016*).

## Comparison to pneumolysin, a bacterial cytolysin

In earlier atomistic MD simulations of PLY (*Vögele et al., 2019*), inserted monomers were ejected from the membrane within 1 µs, unlike what we observed here for GSDMD^NT. Trimers and pentamers of inserted PLY formed narrow pores similar to the small GSDMD^NT pores found here, but with a more circular shape. However, the globular backbone structure of PLY did not crack in the earlier simulations. Both of these observations can likely be explained by the different architectures of PLY and GSDMD. Whereas GSDMD^NT comprises only one domain, PLY is made of four domains connected by disordered loops that act as flexible hinges (*Marshall et al., 2015*; *Lawrence et al., 2015*; *van Pee et al., 2016*). In addition, the β-sheet of PLY is about twice as long as that of GSDMD^NT. These flexible elements likely allow the globular domains 1, 2, and 4 of PLY to stay in tight contact with each other even when the inserted β-sheet is highly curved at its membrane-inserted tip.

In our atomistic MD simulations starting with a lipid-filled GSDMD^NT 33-mer ring, a bicelle-like lipid plug detached from the protein and started to diffuse out of the pore region (*Figure 3*). This is in agreement with observations from AFM experiments, in which material initially filling CDC or gasdermin pores gets cleared over time (*Leung et al., 2014*; *Mulvihill et al., 2018*). While the process of lipid plug expulsion is overall similar to what we saw earlier in coarse-grained simulations of PLY (*Vögele et al., 2019*), there is one notable difference: the lipid plug remained flat during the entire simulation and did not curl up into a small vesicle. This pathway thus adds yet another variation to the alternative routes for lipid escape from the pore seen in MD simulations of cytolysins (*Vögele et al., 2019*; *Vögele et al., 2020*; *Desikan et al., 2020*). The critical diameter $D_c$, above which a small

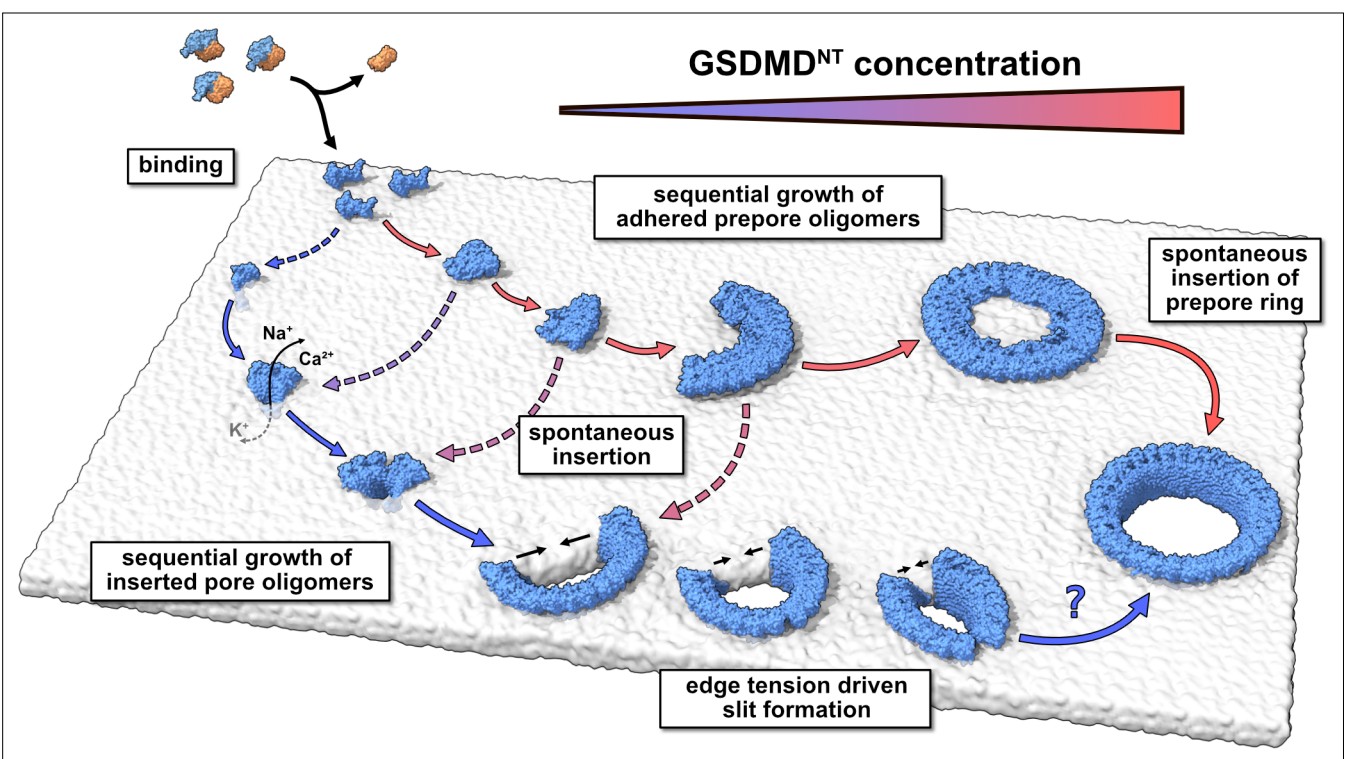

**Figure 6.** Model of membrane pore formation by GSDMD^NT. After proteolytic cleavage, GSDMD^NT monomers bind the inner leaflet of the plasma membrane. Aided by specific lipid interactions they multimerize and, at a critical size, spontaneously insert into the membrane. Depending on the concentration of membrane adhered GSDMD^NT, the insertion may proceed either from a fully formed prepore ring or from small oligomeric assemblies. Dotted arrows indicate that the mechanism of β-sheet insertion so far remains unresolved. Pores formed by small oligomers cause early nonspecific ion flux and can combine with one another or grow sequentially by the attachment of uninserted monomers. Depending on the edge tension in the cellular milieu, arcs would continue to grow or crack to form slit-shaped pores (bottom). Whether slit-shaped pores can grow to circular pores by subsequent mono- or oligomer attachment is unclear.

circular membrane patch becomes mechanically unstable against a transition into a vesicle, satisfies $\pi D_c \gamma = 8\pi\kappa$. For our 20 nm patch to become unstable under the calculated edge tension of $\gamma = 86.4$ pN, the bending rigidity $\kappa$ would have to be below 26 $k_BT$. For our cholesterol-rich membrane, we expect a higher bending rigidity, which would explain why the plug stayed flat.

### The pathway of pore formation is concentration dependent

Taking together all our results, we conclude that GSDMD forms plasma membrane pores following a concentration-dependent kinetic model that is similar to what *Gilbert and Sonnen, 2016* proposed for PLY, but highlights the specifics of GSDMD$^{NT}$ pore assembly (*Figure 6*). At low abundance of prepore GSDMD$^{NT}$ on the membrane, we expect that small oligomers dominate. Eventual membrane insertion would then lead to small sublytic pores (*Figure 4*) that can account for the early ion flux observed experimentally (*Chen et al., 2016*; *de Vasconcelos et al., 2019*; *Rühl and Broz, 2022*) and that could grow further by monomer addition and fusion. By contrast, high abundance of prepore GSDMD$^{NT}$ on the membrane would tilt the kinetic balance towards the assembly of larger prepore assemblies all the way to complete prepore rings and the collective insertion of the full β-sheet. The formation of large prepore oligomers and rings would also be favored on membranes with abundant PI(4,5)P$_2$ based on their preferential incorporation into GSDMD$^{NT}$ interfaces (*Figure 1C*). As a result of the ordering effect of the lipid environment, the enthalpic energy contribution of forming hydrogen bonds during folding, and the energy penalty associated with unformed backbone hydrogen bonds, we expect that folding of the β-hairpins and insertion into the membrane happen in concert. However, due to the complex interplay between lipid dynamics, solvent dynamics, and potentially cooperative hairpin folding, we could not resolve the assembly and membrane insertion process with atomistic MD simulations. The resulting lack of information on the energetic landscape of the insertion process of GSDMD$^{NT}$ and CDCs leaves it open at which oligomer size and local membrane composition insertion becomes kinetically feasible. Resolving this question is a major challenge that will likely require a concerted effort combining structural studies with dynamic imaging experiments and advanced MD simulations that take account also of the cellular context (*Khalid and Rouse, 2020*).

## Materials and methods
### MD simulation parameters

All MD simulations were performed with GROMACS version 2020.3 (*Abraham et al., 2015*) using the CHARMM36m forcefield (*Huang et al., 2017*), the TIP3P water model (*Jorgensen et al., 1983*), and an integration timestep of 2 fs. To calculate electrostatic interactions, we used the Particle Mesh Ewald (PME) algorithm (*Essmann et al., 1995*). Furthermore, we constrained the bond length of heavy atoms to hydrogens using the LINCS algorithm (*Hess et al., 1997*).

During equilibration simulations in the NPT ensemble, a constant pressure was established with a semiisotropic Berendsen barostat (*Berendsen et al., 1984*) with the $x$ and $y$ dimensions of the simulation box coupled together. The reference pressure was 1 bar, the compressibility factor $4.5 \times 10^{-5}$ bar$^{-1}$, and the barostat time constant 5 ps. A constant temperature of 37 °C (310.15 K) was established with three separate Berendsen weak-coupling thermostats (*Berendsen et al., 1984*) applied to the protein, the solvent (including NaCl ions), and the membrane, each with a time constant of 1 ps.

The same general ensemble settings as for the equilibration simulations were used also for all production simulations. However, instead of the Berendsen weak-coupling algorithms we used the Parrinello-Rahman barostat for pressure control (*Parrinello and Rahman, 1981*) and the velocity-rescale algorithm for temperature control (*Bussi et al., 2007*).

Visual analysis as well as image and movie generation were done using VMD (*Humphrey et al., 1996*), PyMOL (*Schrödinger LLC and Warren DeLano, 2015*), and ChimeraX (*Pettersen et al., 2021*).

### Membrane setup

To simulate GSDMD$^{NT}$ in its native environment, we designed a plasma membrane mimetic with asymmetric lipid composition. Following our earlier work (*Schaefer et al., 2021*) and *Lorent et al., 2020*, the outer leaflet of our membrane is rich in sphingolipid and phospholipids with PC headgroups, and holds very few poly-unsaturated lipids. By contrast, the inner leaflet has a high content of lipids with PE headgroups and mostly polyunsaturated tails. In addition, the inner leaflet contains negatively

**Table 3.** Asymmetric plasma membrane composition.
Fatty acid tails abbreviated as FA.

| Lipid | FA | Full name | Charge | Inner leaflet [mol %] | Outer leaflet [mol %] |
|---|---|---|---|---|---|
| CHOL | | Cholesterol | 0 | 40.4 | 40.0 |
| PSM | 18:1/16:0 | N-palmitoyl-D-erythro-sphingosylphosphorylcholine | 0 | 1.0 | 12.0 |
| NSM | 18:1/24:1 | N-nervonoyl-D-oleoyl-sphingosylphosphorylcholine | 0 | - | 9.0 |
| LSM | 18:1/24:0 | N-lignoceroyl-D-oleoyl-sphingosylphosphorylcholine | 0 | - | 8.0 |
| PLPC | 16:0/18:2 | 1-palmitoyl-2-linoleoyl-sn-glycero-3-phosphocholine | 0 | 8.1 | 15.0 |
| SOPC | 18:0/18:1 | 1-stearoyl-2-oleoylphosphocholine | 0 | - | 7.0 |
| PAPC | 16:0/20:4 | 1-palmitoyl-2-arachidonoyl-glycero-3-phosphocholine | 0 | - | 5.0 |
| POPC | 16:0/18:1 | 1-palmitoyl-2-oleoyl-glycero-3-phosphocholine | 0 | 3.0 | - |
| DPPC | 16:0/16:0 | 1,2-dipalmitoyl-glycero-3-phosphocholine | 0 | 2.0 | - |
| PLA20(PE) | 18:0/20:4 | 1-O-stearoyl-2-O-arachidonoyl-glycero-3-phosphoethanolamine | 0 | 11.1 | 3.0 |
| PDoPE | 16:0/22:6 | 1-palmitoyl-2-docosahexaenoyl-glycero-3-phosphoethanolamine | 0 | 8.1 | - |
| SAPE | 18:0/20:4 | 1-stearoyl-2-arachidonoyl-glycero-3-phosphoethanolamine | 0 | 4.0 | - |
| POPE | 16:0/18:1 | 1-palmitoyl-2-oleoyl-glycero-3-phosphoethanolamine | 0 | 3.0 | - |
| PAPS | 16:0/20:4 | 1-palmitoyl-2-arachidonoyl-glycero-3-phosphoserine | -1 | 13.1 | - |
| SAPS | 18:0/20:4 | 1-stearoyl-2-arachidonoyl-glycero-3-phosphoserine | -1 | 1.0 | 1.0 |
| PI(4,5)P$_2$ | 16:0/18:2 | 1-palmitoyl-2-linoleoyl-sn-glycero-3-phosphoinositol-4,5-bisphosphate | -4 | 5.1 | - |

charged PI(4,5)P$_2$ and PS lipids. Both leaflets were built with the same cholesterol concentration of 40 mol % (**Lorent et al., 2020**). The full composition is summarized in **Table 3**.

A small membrane patch with 99 lipids in the inner leaflet and 100 lipids in the outer leaflet, surrounded by water and 150 mM NaCl, was created using the Charmm-GUI membrane builder (**Jo et al., 2007**; **Jo et al., 2008**). This membrane system was energy minimized using a steepest descent algorithm until the highest force acting on any atom fell below 1000 kJ mol$^{-1}$. The minimized system was then temperature and pressure equilibrated in six MD simulation steps with decreasing restraints on the phosphate headgroups and lipid tail dihedral angles (**Table 4**).

Finally, the membrane was MD simulated for 5 μs at 70 °C to promote cholesterol flip-flop that allows the membrane to equilibrate to tensionless leaflets (**Miettinen and Lipowsky, 2019**). After 3.5

**Table 4.** Restraints used during energy minimization (EM) and stepwise equilibration (steps 1–6) of the asymmetric plasma membrane mimetic in kJ mol$^{-1}$nm$^{-2}$.

| Step | Time [ns] | Timestep [fs] | Ensemble | Lipids | Dihedrals |
|---|---|---|---|---|---|
| EM | | | | 1000 | 1000 |
| 1 | 0.125 | 1 | NVT | 1000 | 1000 |
| 2 | 0.125 | 1 | NVT | 400 | 400 |
| 3 | 0.125 | 1 | NPT | 400 | 200 |
| 4 | 0.5 | 2 | NPT | 200 | 200 |
| 5 | 0.5 | 2 | NPT | 40 | 100 |
| 6 | 0.5 | 2 | NPT | 0 | 0 |

μs no further net cholesterol flip-flop was observed and the resulting membrane patch was used to assemble all larger membranes used in this work.

## Membrane edge tension simulation

To estimate the line tension of an open membrane edge of our plasma membrane composition, we removed a 9 nm wide strip of lipids from a 20×20 nm² membrane patch in the $xy$ plane. After resolvating the simulation box and deleting water molecules that were positioned between the membrane headgroup planes, this open membrane system was energy minimized and equilibrated for 5 ns (using parameters EM and Step 6 from *Table 3*). Following *Jiang et al., 2004*, between these two steps we rotated the entire system around the $y$-axis by 90 degrees to orientate the open membrane edge along $z$. This allowed us to fix the length of the box along $z$ and therefore the length of the open edge, while pressure coupling the remaining two dimensions together to a target value of 1 bar. We then simulated the open membrane edge system for 530 ns. From this simulation, we then calculated the membrane edge tension $\gamma$ by averaging the difference in lateral and normal pressures according to *Jiang et al., 2004*

$$\gamma = \tfrac{1}{2} \left\langle L_x L_y \left[ \tfrac{1}{2}(P_{xx} + P_{yy}) - P_{zz} \right] \right\rangle \tag{1}$$

with $L_x$ and $L_y$ being the fluctuating box dimensions in the $x$ and $y$ directions, respectively, and $P_{xx}$, $P_{yy}$, and $P_{zz}$ being the diagonal elements of the pressure tensor. The factor 1/2 accounts for the fact that we have two open membrane edges in our simulation system. Only the last 500 ns of the production run were considered for calculating the edge tension.

## Pore conformation monomer and oligomer system setup

Using Charmm-GUI (*Jo et al., 2007*; *Jo et al., 2008*), we extracted one monomer from the GSDMD[NT] structure in pore conformation (PDB Id 6VFE [*Xia et al., 2021*]), reversed the L192E mutation back to its native sequence, and added capping groups to the termini to mimic the continuation of the peptide bond (acetylated N-terminus, amidated C-terminus). To build higher level multimers, we then copied the monomer as often as necessary and aligned it with the next respective subunit of the complete ring structure. In this way, we built GSDMD[NT] 2-mer, 3-mer, 5-mer, 8-mer, 10-mer, 16-mer, and 27-mer arcs, as well as a full ring consisting of 33 subunits.

The resulting monomer and the multimeric arc and ring structures were inserted into membranes created by replicating the previously described equilibrated membrane patch in the $x$ and $y$ directions. To remove clashes of overlaying lipid and protein atoms, all lipid residues with at least one atom closer than 2 Å to any protein atom were removed. Additionally, lipids from within the pore were also removed in the case of the full ring system. To counteract the asymmetry introduced by removing different numbers of lipids from the outer and inner leaflet, we removed additional lipids from the leaflet containing excess lipids. We treated cholesterol molecules and all other phospholipids separately, and for each excess lipid of the respective group we removed a random lipid out of the leaflet holding more lipids.

Subsequently, all systems were solvated and ions were added. Due to the high negative charge of the membrane (in particular the inner leaflet) we introduced 150 mM of sodium ions to the system and then added excess chloride ions to neutralize the systems. All systems were then energy minimized as

**Table 5.** Restraints used during energy minimization (EM) and stepwise equilibration (1–3) of GSDMD[NT] pore conformation systems in kJ mol⁻¹nm⁻².

| Step | Time [ns] | Timestep [fs] | Backbone | Sidechains | Lipid headgroup ($z$) | Water ($z$) |
|------|-----------|---------------|----------|------------|----------------------|-------------|
| EM   |           |               | 4000     | 2000       | 1000                 | 1000        |
| 1    | 5*        | 2             | 2000     | 1000       | 10                   | 50*         |
| 2    | 50        | 2             | 2000     | 1000       | 50                   | 0           |
| 3    | 80        | 2             | 500      | 100        | 0                    | 0           |

*For the 27-mer system with an initially lipid filled pore this was extended to 10 ns with a force constant of 1000 kJ mol⁻¹ nm⁻² for restraining the z-position of water molecules.

**Table 6.** Atomistic GSDMD$^{NT}$ simulations.

| Conformation | Subunits | Simulated time [μs] | Membrane size [nm²] | No. of atoms |
|---|---|---|---|---|
| prepore | 1 | 7.0 | 13.4×13.4 | 222740 |
| prepore | 33 | 3.5 | 38.9×38.9 | 1950552 |
| *prepore | 3, 5, 16 | 1.5 | 38.9×38.9 | 1916829 |
| prepore | 33 | 2.2 | 45.8×45.8 | 3113594 |
| pore | 1 | 5.0 | 12.7×12.7 | 216375 |
| pore | 2 | 5.0 | 12.5×12.5 | 213665 |
| pore | 3 | 5.0 | 12.2×12.2 | 208656 |
| pore | 5 | 4.3 | 18.8×18.8 | 481242 |
| pore | 8 | 2.5 | 25.5×31.9 | 1079514 |
| pore | 10 | 3.0 | 25.4×31.7 | 1071088 |
| pore | 16 | 3.5 | 31.3×31.3 | 1329398 |
| †pore | 16 | 1.4 | 34.9×34.9 | 1497819 |
| pore | 27 | 4.1 | 37.2×37.2 | 2226909 |
| pore | 33 | 5.0 | 37.0×37.0 | 2119785 |
| ‡pore | 33 | 1.5 | 37.0×37.0 | 2119785 |
| §pore | 33 | 1.5 | 37.0×37.0 | 1885031 |

*simulated cut out oligomers of the full 33-mer prepore ring after 200 ns.
†simulated in pure DOPC membrane.
‡70°C continuation of the 37°C simulation after 5 μs.
§simulated with lipid plug inside pore.

described above for the membrane system and subsequently equilibrated in three steps with varying restraints (*Table 5*). Note that unlike the equilibration of the membrane-only system, here lipid tail dihedral angles were never restrained, to allow for fast filling of space freed up by the removal of excess lipids. After equilibration, all systems were simulated without restraints and using the above described production simulation parameters for the simulation times summarized in *Table 6*.

## Prepore conformation mono- and oligomer system setup

For setting up systems with GSDMD$^{NT}$ in prepore conformation, we took the back-mutated monomer that we used for the pore conformation simulations and manually bent its β-hairpins out of the membrane using the structure editing tool of PyMOL (*Schrödinger LLC and Warren DeLano, 2015*). We then set up the simulation box setup and performed an energy minimization as for the systems in pore conformation. In the following 10 ns long equilibration run, only the positions of heavy protein atoms were restrained with force constants of 500 (backbone) and 100 (sidechains) kJ mol$^{-1}$ nm$^{-2}$, respectively. From the simulation of the prepore 33-mer, we extracted the structure after 200 ns and removed three times three consecutive subunits from the ring assembly in a way that a 3-mer, a 5-mer and a 16-mer remained in the system. The resulting system was then simulated alongside the original 33-mer prepore simulation.

## Lipid binding site analysis

Using the PyLipID python library (*Song et al., 2022*) and own code, we analyzed protein-lipid interactions for the prepore monomer and the full ring system in pore conformation. In particular, we characterized the interactions of the headgroup of PI(4,5)P$_2$ and PS lipids with the protein. Here, we disregarded the first 500 ns of the production simulation and, from the remaining frames, collected all interactions where any headgroup atom came closer than 3.6 Å with respect to any protein residue. In case of the full 33-mer pore, we averaged lipid interactions over all 33 subunits. To

minimize 'rattling-in-a-cage' effects, we made use of the dual cutoff scheme suggested for PyLipID when analyzing the duration of these contacts. In this case, the duration of a lipid-protein contact was counted until the distance exceeded 5 Å. With this setup, assisted by visual analysis, we identified representative $PI(4,5)P_2$ binding sites. Counting of lipid-protein interactions was done on the full trajectories, with the same 3.6 Å cutoff and only counting interactions between headgroup heavy atoms and protein heavy atoms.

## Acknowledgements

We thank Jürgen Köfinger, Hendrik Jung, Balázs Fábián, Shanlin Rao, and Sergio Cruz-León for helpful discussions. This work was supported by the Max Planck Society, the Clusterproject ENABLE funded by the Hessian Ministry for Science and the Arts, and the Collaborative Research Center 1507 funded by the Deutsche Forschungsgemeinschaft (DFG, German Research Foundation). We also thank the Max Planck Computing and Data Facility (MPCDF) for computational resources.

## Additional information

### Funding

| Funder | Grant reference number | Author |
| --- | --- | --- |
| Deutsche Forschungsgemeinschaft | CRC 1507 | Stefan L Schaefer Gerhard Hummer |
| Max-Planck-Gesellschaft | | Stefan L Schaefer Gerhard Hummer |
| Hessisches Ministerium für Wissenschaft und Kunst | Cluster project ENABLE | Stefan L Schaefer Gerhard Hummer |

The funders had no role in study design, data collection and interpretation, or the decision to submit the work for publication.

### Author contributions

Stefan L Schaefer, Data curation, Software, Formal analysis, Validation, Investigation, Visualization, Methodology, Writing - original draft, Writing - review and editing; Gerhard Hummer, Conceptualization, Resources, Supervision, Funding acquisition, Validation, Methodology, Writing - original draft, Project administration, Writing - review and editing

### Author ORCIDs

Stefan L Schaefer  http://orcid.org/0000-0001-7942-8701
Gerhard Hummer  http://orcid.org/0000-0001-7768-746X

### Decision letter and Author response

Decision letter https://doi.org/10.7554/eLife.81432.sa1
Author response https://doi.org/10.7554/eLife.81432.sa2

## Additional files

### Supplementary files

• MDAR checklist

### Data availability

The data in this study are included in the manuscript and source data files. Analysis code, raw trajectories and simulation parameter files are deposited at DOI: https://doi.org/10.5281/zenodo.6797842.

The following dataset was generated:

| Author(s) | Year | Dataset title | Dataset URL | Database and Identifier |
|---|---|---|---|---|
| Schaefer SL, Hummer G | 2022 | Raw data for: Sublytic gasdermin-D pores captured in atomistic molecular simulations | https://doi.org/10.5281/zenodo.6797842 | Zenodo, 10.5281/zenodo.6797842 |

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
