## [Editor Report]

This article will be of interest to cell biologists, structural biologists, and biophysicists studying programmed cell death, membrane transport, and protein-lipid interactions. The simulation data presented offers atomistic detail of how gasdermin-D N-terminal domains assemble on the plasma membrane and trigger the formation of membrane pores which lead to pyroptosis. The study is well-designed and the resulting data are rigorously analyzed.

---

## [Decision Letter]

**Decision letter after peer review:**

Thank you for submitting your article "Sublytic gasdermin-D pores captured in atomistic molecular simulations" for consideration by *eLife*. Your article has been reviewed by 2 peer reviewers, and the evaluation has been overseen by José Faraldo-Gómez as the Senior Editor. The reviewers have opted to remain anonymous.

The reviewers have discussed their reviews with one another, and the Senior Editor has compiled this letter to help you prepare a revised submission. Reviewers and Editors concur your manuscript is potentially acceptable for publication in *eLife*. A final decision on publication will however require a compelling response to the various queries raised below.

*Reviewer #1 (Recommendations for the authors):*

(1) To provide binding data for other lipids in the membrane, as a supplement for Figure 1, for comparison – if only as a null result – with the PI(4,5)P2 results.

(2) To put membrane deformation due to gasdermin prepores in context of similar (experimental) observations made for cholesterol dependent cytolysins suilysin (Leung et al., 2014, cited in manuscript) and pneumolysin (Faraj et al., Sci Rep 2020).

(3) To provide evidence that smaller oligomers are not only stable when inserted in the membrane, but that such oligomers can also overcome any energy barrier involved in transiting from prepore to pore state.

(4) For comparison with the arc-shaped assemblies in Figure 4, to include data that demonstrate the fate of the lipids when the membrane is locally enclosed by a full, ring-shaped gasdermin assembly (as in Figure 2C).

(5) To show multiple MD simulations of differently sized larger (>10) arc-shaped assemblies, to identify how common the "cracking" of the arc assemblies occurs, as well as possibly defining its size dependence.

(6) To compare the gasdermin-D results in greater detail with previous MD simulations of pneumolysin assemblies, which form similar β-barrel pores but appear less prone to cracking.

(7) To adjust / refine conclusions re different pathways as identified in Figure 5, clarifying more explicitly how the MD results sustanstiate the selection of these two different pathways, and why membrane insertion of larger arc-shaped prepores is excluded as a possible pathway.

*Reviewer #2 (Recommendations for the authors):*

It is appropriate for publication in *eLife*. I only have one minor concern that relates to the author's proposed model to be addressed.

1. The authors used a cryo-EM structure (PDB ID: 6VFE) obtained in liposomes around pH 7 (Xia S., et al., Nature, 2021, 593 (7860): 607-611). This 33-mer forms an intact transmembrane β-barrel comprised of 66 β-hairpins (2 per monomer). However, there is one GSDMDNT structure crystallized at pH 5.5 (PDB ID: 6N9O) that has these β-hairpins mostly unresolved (Liu Z., et al., Immunity, 2019, 51 (1): 43-49), indicating that the region is flexible and unfolded under crystallization conditions. This discrepancy raises a couple of questions.

1) How may pH impact the conformation, especially the β-hairpin region of GSDMDNT? Are different conformations of this region seen in the two PDB structures due to varying experimental pH values?

2) If not, this discrepancy suggests that either other GSDMDNT monomers or membranes stabilize the β-hairpins:

A) In the prepore monomer and 33-mer simulations, these β-hairpins at the membrane surface are unfolded (Figure 1A). This suggests that it is more likely the membranes that stabilize these β-hairpins. Note there is no high-resolution prepore structure available. The authors prepared a prepore starting structure from the pore conformation by bending these β-hairpins out of the membrane. Did these β-hairpins automatically unfold in the prepore simulations?

B) It seems to me that the prepore 33-mer and pore 33-mer simulations (Figure 2) are two independent simulations. If yes to point A), have the authors ever tried to model how a prepore 33-mer would insert into the membranes and mature into a pore 33-mer with these β-hairpins folded back, starting from an equilibrated prepore 33-mer structure (i.e., these β-hairpins are gone)? Or have the authors tried the opposite transition, i.e., from a pore 33-mer to prepore 33-mer?

C) Alternatively, have the authors tried the monomer case, i.e., modeled how a GSDMDNT monomer would insert its unfolded β-hairpins into the membranes and fold back? It looks like the all the pore simulations (from monomer to 27-mer, Figure 3) were initiated from a membrane-inserted GSDMDNT structure with folded β-hairpins.

Having said this, I am wondering if there have been any studies that demonstrate how these β-hairpins fold during the membrane insertion. Note that I am not challenging the authors' simulations or conclusions. I am only worried that the proposed model (Figure 5) is missing a certain element.

---

## [Author Response]

Reviewer #1 (Recommendations for the authors):(1) To provide binding data for other lipids in the membrane, as a supplement for Figure 1, for comparison – if only as a null result – with the PI(4,5)P2 results.

We thank the reviewer for the overall very positive assessment and for pointing out the lack of comparison across lipid species. In response, in Figure 1—figure supplement 1, we now show the number of lipids, resolved by head group type, in contact with the 33-mer gasdermin ring as a function of time. In this plot, one can see that both PI(4,5)P_2_ and PS lipids bind preferentially to GSDMD^NT^ over zwitterionic lipid types and cholesterol. In terms of relative numbers, the binding preference to PI(4,5)P_2_ lipids is the strongest.

(2) To put membrane deformation due to gasdermin prepores in context of similar (experimental) observations made for cholesterol dependent cytolysins suilysin (Leung et al., 2014, cited in manuscript) and pneumolysin (Faraj et al., Sci Rep 2020).

We thank the reviewer for pointing us to additional references on membrane binding in the prepore state. We now cite the listed papers and discuss our results in context with their findings. We now write:

“Similar to the recruitment of PI(4,5)P2 by GSDMD^NT^, prepore oligomers of pneumolysin (PLY), a cholesterol dependent cytolysin (CDC), have been shown to recruit cholesterol, which locally enhanced lipid order (Faraj et al., 2020).”

We now also discuss Leung et al. (2014) in the context of the insertion of smaller oligomers, arc to ring growth, and the ejection of the lipid plug:

“Combined with our observation of stable prepore oligomers and the identification of stable, presumably inserted, arcs of the CDC suilysin comprising as few as 5 subunits (Leung et al. 2014), this finding opens up the possibility that GSDMD^NT^ inserts its β-sheet already from such small oligomers. In contrast to PLY, GSDMD^NT^ was shown to permit growth of inserted arcs (Mulvihill et al., 2018) and we believe that such stably inserted small oligomers could diffuse together to form slit and ringshaped pores (Leung et al., 2014).” and “This is in agreement with observations from AFM experiments, in which material initially filling CDC or gasdermin pores gets cleared over time (Leung et al., 2014, Mulvihill et al. 2018).”

(3) To provide evidence that smaller oligomers are not only stable when inserted in the membrane, but that such oligomers can also overcome any energy barrier involved in transiting from prepore to pore state.

We agree with the reviewer that at present we do not have information on the initial insertion process and the associated barriers from experiment or simulation. In response, we now make this clearer in our manuscript. We emphasize that we currently do not know at which oligomer size insertions become kinetically feasible. We also changed the statement to oligomers “supporting” membrane pores instead of “creating” them.

At present, we have only a vague understanding of the actual insertion process and the associated energetic barriers, from experiment or simulation. We now make this clear in the writing and highlight the poorly understood question of membrane insertion as a major emerging challenge both for gasdermins and bacterial cytolysins. We now write:

“However, due to the complex interplay between lipid dynamics, solvent dynamics, and potentially cooperative hairpin folding, we could not resolve the assembly and membrane insertion process with atomistic MD simulations. The resulting lack of information on the energetic landscape of the insertion process of GSDMD^NT^ and CDCs leaves it open at which oligomer size and local membrane composition insertion becomes kinetically feasible. Resolving this question is a major challenge that will likely require a concerted effort combining structural studies with dynamic imaging experiments and advanced MD simulations that take account also of the cellular context (Khalid and Rouse, 2020).”

(4) For comparison with the arc-shaped assemblies in Figure 4, to include data that demonstrate the fate of the lipids when the membrane is locally enclosed by a full, ring-shaped gasdermin assembly (as in Figure 2C).

We thank the reviewer for suggesting this very interesting simulation system. In response we included the description of a simulation of a 33-mer ring in pore conformation with an initially entrapped lipid plug. In the text and in the newly added main text Figure 3 we now describe how the trapped membrane quickly retracts from the hydrophilic inside of the gasdermin pore and forms a bicelle that starts to escape from the pore. In the new section on “Comparison to pneumolysin, a bacterial cytolysin”, we also compare and contrast this process to what was reported before for pneumolysin and other cytolysins.

(5) To show multiple MD simulations of differently sized larger (>10) arc-shaped assemblies, to identify how common the "cracking" of the arc assemblies occurs, as well as possibly defining its size dependence.

In response to the criticism that we only reported a single (N=1) cracking event, we now added additional trajectories showing such cracking events to Figure 4 — figure supplement 1. We now show that oligomers of size three or smaller are stable, and oligomers of size five and larger eventually all crack. We also now make it clear that we observed cracking twice for large arcs (size 16, once in the plasma membrane and once in the pure DOPC membrane). To make this clear to the reader, we now also added a table to the main text, in which we summarize the cracking of arcs in our simulations (Table 1), and we included triangles that indicate arc cracking to all snapshots in which arc cracking is visible (see Figures 2 — figure supplement 1, Figure 4, Figure 5 and Figure 5 — figure supplement 1).

(6) To compare the gasdermin-D results in greater detail with previous MD simulations of pneumolysin assemblies, which form similar β-barrel pores but appear less prone to cracking.

We thank the reviewer for inviting us to discuss in depth how our results for gasdermin relate to the earlier studies of pneumolysin. In response, we added a paragraph in which we compare the two systems entitled “Comparison to pneumolysin, a bacterial cytolysin”. In this paragraph we describe the similarities and differences we observe in a comparison to our previous simulations of pneumolysin and the simulations presented in this work. Where possible, we also relate these differences to their different architecture, in particular in an effort to explain the different propensities of PLY and GSDMD^NT^ rings to crack.

(7) To adjust / refine conclusions re different pathways as identified in Figure 5, clarifying more explicitly how the MD results sustanstiate the selection of these two different pathways, and why membrane insertion of larger arc-shaped prepores is excluded as a possible pathway.

We thank the reviewer for alerting us of this issue. In response we now include the description of a simulation of additional prepore assemblies (3-mer, 5-mer, 16-mer). We describe their stability, the mild membrane deformation caused by the 16-mer and visualize them in Figure 2 —figure supplement 1. In addition, we included this additional pathway as one of the options in the model and reworked Figure 6 (previously Figure 5).

Reviewer #2 (Recommendations for the authors):It is appropriate for publication in eLife. I only have one minor concern that relates to the author's proposed model to be addressed.1. The authors used a cryo-EM structure (PDB ID: 6VFE) obtained in liposomes around pH 7 (Xia S., et al., Nature, 2021, 593 (7860): 607-611). This 33-mer forms an intact transmembrane β-barrel comprised of 66 β-hairpins (2 per monomer). However, there is one GSDMDNT structure crystallized at pH 5.5 (PDB ID: 6N9O) that has these β-hairpins mostly unresolved (Liu Z., et al., Immunity, 2019, 51 (1): 43-49), indicating that the region is flexible and unfolded under crystallization conditions. This discrepancy raises a couple of questions.1) How may pH impact the conformation, especially the β-hairpin region of GSDMDNT? Are different conformations of this region seen in the two PDB structures due to varying experimental pH values?

We thank the reviewer for the overall very positive assessment of our manuscript.

Indeed, the lack of experimental insight into the structure of the (to be) β-hairpins outside of the pore conformation is striking. Human GSDMD comprises three histidine residues in the unresolved region of the crystal structure that could be susceptible to (de-)protonation in the discussed pH range. It is conceivable that especially His176, which is surrounded by basic residues, would disturb the local ordering at low pH. However, the low resolution of the hairpins is not only a problem of the mentioned structure (6N9O), but in fact of all non-pore structures of any eukaryotic gasdermin. This includes the additional mouse GSDMA3 (5B5R, crystallized at pH 6.5; Ding, J. et al., Nature 535, 111–116 (2016)) and mouse GSDMD (6N9N, crystallized at pH 5.5; Liu Z., et al., Immunity, 2019, 51 (1): 43-49) structures. Importantly, it also includes the density of the pre-pore found on top of the resolved pore structures Xia et al. report (pH 7).

Given that in all non-pore structures (including at pH 7) the resolution of the prepore hairpins is low, it is tempting to speculate that the observed disorder in nonpore structures reflects significant disorder outside the membrane, with the order increasing once the hairpins are (partially) inserted. We included a statement that clarifies our expectations about when folding of the β-hairpins occurs to the manuscript:

“As a result of the ordering effect of the lipid environment, the enthalpic energy contribution of forming hydrogen bonds during folding, and the energy penalty associated with unformed backbone hydrogen bonds, we expect that folding of the β-hairpins and insertion into the membrane happen in concert."

We note further that uninserted β-hairpins are unstable and unfold in our simulations, as discussed in response to the following question (point 2A).

2) If not, this discrepancy suggests that either other GSDMDNT monomers or membranes stabilize the β-hairpins:A) In the prepore monomer and 33-mer simulations, these β-hairpins at the membrane surface are unfolded (Figure 1A). This suggests that it is more likely the membranes that stabilize these β-hairpins. Note there is no high-resolution prepore structure available. The authors prepared a prepore starting structure from the pore conformation by bending these β-hairpins out of the membrane. Did these β-hairpins automatically unfold in the prepore simulations?

Yes, in the simulation of the pre-pore structure, large parts of the initially folded βhairpins unfolded quite quickly in the beginning of the simulation. This finding is consistent with the point of the reviewer, and is now discussed. We now write:

“Within the first nanoseconds of the production simulation, its originally folded βhairpins unfolded and the ring deformed the membrane upwards into a crown shape (Figure 2B).”

B) It seems to me that the prepore 33-mer and pore 33-mer simulations (Figure 2) are two independent simulations. If yes to point A), have the authors ever tried to model how a prepore 33-mer would insert into the membranes and mature into a pore 33-mer with these β-hairpins folded back, starting from an equilibrated prepore 33-mer structure (i.e., these β-hairpins are gone)? Or have the authors tried the opposite transition, i.e., from a pore 33-mer to prepore 33-mer?

We thank the reviewer for raising this interesting point. Yes, the prepore 33-mer and pore 33-mer simulations are different simulations, which we now make clearer:

“we performed simulations of the full 33-mer assemblies, starting separate simulations from its proposed prepore conformation and from its resolved pore conformation (Xia et al., 2021).”

On a molecular level the mechanism of insertion is currently poorly understood and likely involves a complex interplay between (potentially cooperative) protein folding and lipid and solvent dynamics. While methods exist to bias simulations so that insertion (or extraction) would occur, these methods require long run times and many replicates to cover the whole conformational space to produce meaningful insight. In our view, for systems of this complexity and size, molecular dynamics simulations covering the insertion process would at present be too speculative.

C) Alternatively, have the authors tried the monomer case, i.e., modeled how a GSDMDNT monomer would insert its unfolded β-hairpins into the membranes and fold back? It looks like the all the pore simulations (from monomer to 27-mer, Figure 3) were initiated from a membrane-inserted GSDMDNT structure with folded β-hairpins.

The above considerations are even true for relatively smaller systems comprising a single monomer of GSDMD (~220,000 atoms). In our opinion, the two crucial bottlenecks are the slow lipid dynamics and protein folding that we expect to happen alongside the insertion. It is known that the dynamics of membrane insertion even of small molecules are prone to sampling errors. They emerge from the selection of starting structures as well as from long lasting (>1 µs) metastable states and slow lipid dynamics (Neale et al. BBA Biomembranes, 2016, 2539-2548; You et al., J. Chem. Theory Comput. 2019, 15, 4, 2433–2443). Overcoming these challenges would require extensive and sophisticated enhanced sampling. Protein folding in molecular dynamics has been done before; however, even the folding of small peptides in aqueous medium requires simulation times in the tens of µs. We therefore expect that a meaningful elucidation of the mechanism of insertion of even one individual monomer, with the current biasing techniques, would require immense computational resources and effort that are beyond the scope of this publication.

We again thank the reviewer for alerting us that this was not explicitly stated in the manuscript. In response we added a brief explanation of the underlying reasons for why we did not include simulations targeting the mechanism of insertion to the concluding subsection, just before Materials and Methods.

Having said this, I am wondering if there have been any studies that demonstrate how these β-hairpins fold during the membrane insertion. Note that I am not challenging the authors' simulations or conclusions. I am only worried that the proposed model (Figure 5) is missing a certain element.

We thank the reviewer for pointing us towards this potential weakness in our proposed model. We updated the model in Figure 6 to emphasize the remaining uncertainties about the mechanism of hairpin insertion and folding. In the caption of the updated Figure 6, we now write: “Dotted arrows indicate that the mechanism of β-sheet insertion so far remains unresolved.”